# Fatty Liver and Hyperuricemia in Workers: Combined Effects on Metabolic Dysfunction and the Role of Lifestyle Factors

**DOI:** 10.3390/metabo15050318

**Published:** 2025-05-09

**Authors:** Jui-Hua Huang, Ren-Hau Li, Hon-Ke Sia, Feng-Cheng Tang

**Affiliations:** 1Department of Golden-Ager Industry Management, Chaoyang University of Technology, Taichung 413, Taiwan; juihua316@cyut.edu.tw; 2Department of Psychology, Chung Shan Medical University, Taichung 402, Taiwan; davidrh@csmu.edu.tw; 3Room of Clinical Psychology, Chung Shan Medical University Hospital, Taichung 402, Taiwan; 4Division of Endocrinology and Metabolism, Changhua Christian Hospital, Changhua 500, Taiwan; 90279@cch.org.tw; 5Department of Post-Baccalaureate Medicine, College of Medicine, National Chung Hsing University, Taichung 402, Taiwan; 6Graduate Institute of Clinical Medicine, College of Medicine, National Chung Hsing University, Taichung 402, Taiwan; 7Department of Occupational Medicine, Changhua Christian Hospital, Changhua 500, Taiwan; 8Department of Leisure Services Management, Chaoyang University of Technology, Taichung 413, Taiwan

**Keywords:** fatty liver, hyperuricemia, lifestyle, metabolic syndrome, inflammation, liver dysfunction

## Abstract

**Background/Objectives**: Fatty liver and hyperuricemia are growing public health concerns linked to unhealthy lifestyles, yet their combined effects in working populations remain underexplored. This study investigates their associations with metabolic risk factors, inflammation, and liver dysfunction to inform workplace health strategies. **Methods**: The participants were employees aged 20 or older from four industrial enterprises located in central Taiwan. A total of 3089 participants (2571 males, 518 females) were analyzed. Lifestyle factors were assessed via a self-administered questionnaire, fatty liver was diagnosed using ultrasound, and serum uric acid levels, metabolic parameters, inflammatory markers, and liver function were measured. **Results**: The prevalence of fatty liver (43.2%) exceeded that of hyperuricemia (25.5%), with a higher burden among males. Fatty liver was associated with lower physical activity, while alcohol consumption was significantly higher in individuals with both conditions. Both conditions correlated with increased metabolic risk factors, liver dysfunction, and inflammation. Health-related risk factors were compared across four groups, using Group A (no hyperuricemia/no fatty liver, OR = 1.00) as the reference. The risk of metabolic syndrome increased progressively: 2.90 (Group B: hyperuricemia/no fatty liver), 6.15 (Group C: no hyperuricemia/fatty liver), and 11.52 (Group D: hyperuricemia/fatty liver), following the trend A < B < C < D. Notably, Group D had the highest risk, with exacerbated inflammation and liver dysfunction. **Conclusions**: Fatty liver and hyperuricemia synergistically worsen metabolic disorders, inflammation, and liver dysfunction. Early detection and lifestyle interventions are crucial to mitigating long-term health risks.

## 1. Introduction

Fatty liver (FL) and hyperuricemia are increasingly prevalent metabolic disorders that pose significant public health challenges [1,2]. Both conditions are strongly associated with unhealthy lifestyle habits, such as poor dietary choices, physical inactivity, smoking, and excessive alcohol consumption [3,4,5]. Central to their impact is their connection to metabolic syndrome (MS)—a cluster of risk factors that includes central obesity, hypertension, dyslipidemia, and hyperglycemia—all of which contribute to systemic inflammation and elevated cardiovascular risks [6]. Despite advancements in understanding their pathophysiology, significant knowledge gaps persist regarding effective interventions, particularly in working populations, where occupational stress, irregular meal patterns, and sedentary behavior exacerbate these risks [7,8]. Addressing these gaps is vital for reducing the burden of FL and hyperuricemia in this demographic.

Lifestyle habits play a pivotal role in shaping metabolic health outcomes. Physical inactivity, excessive alcohol consumption, and poor dietary habits are strongly associated with the development of FL and hyperuricemia [3,4,5]. For example, sedentary behavior in the workplace has been shown to significantly increase the risk of FL, emphasizing the crucial influence of occupational environments on metabolic health [9]. Interventions that focus on improving dietary habits and promoting regular physical activity tailored to specific job types have demonstrated potential in reducing the prevalence of FL and its associated complications [10]. These findings highlight the need for workplace wellness programs that address both individual behaviors and organizational policies to help mitigate the prevalence of metabolic disorders among employees.

FL is characterized by excessive triglyceride accumulation in the liver, often resulting from obesity, insulin resistance, and poor dietary habits [3]. Similarly, hyperuricemia, defined by elevated serum uric acid levels, is linked to high-purine diets and alcohol consumption [4]. These conditions often coexist, sharing common risk factors such as systemic inflammation and metabolic dysregulation [6,11,12]. Serum uric acid has been identified as an independent predictor of advanced NAFLD, including histological inflammation and fibrosis [13,14]. Recent evidence also highlights the role of serum uric acid as a biomarker for disease progression in FL, with hyperuricemia increasing the risk of significant liver fibrosis in FL patients but not in those without the condition [15]. This interplay underscores the importance of a comprehensive assessment of metabolic and inflammatory markers to guide prevention and intervention strategies tailored to high-risk populations, including workers.

While FL, hyperuricemia, and lifestyle habits have been extensively studied, most research focuses on the general population or high-risk groups (e.g., individuals with obesity or diabetes) [6,16]. Studies on working populations remain limited, despite their exposure to occupational risk factors such as sedentary behavior, poor diet, and job-related stress—factors that may increase FL and hyperuricemia risk. Moreover, existing research typically examines these conditions separately, overlooking their combined effects on metabolic health, inflammation, and liver function. This study addresses these gaps by investigating lifestyle influences on FL and hyperuricemia in workers and analyzing their coexistence’s impact on metabolic health and liver function. The findings will inform workplace health programs to reduce metabolic disorders, enhance well-being, and improve productivity.

## 2. Materials and Methods

### 2.1. Ethical Approval

Under Taiwan’s Labor Health Protection Rule, occupational health professionals must regularly assess employees’ health risks and needs. As part of the Taiwan Workplace Health Promotion Scheme, this study utilized public health monitoring among employees. Participants received an information sheet explaining this study alongside the questionnaire. Participation was voluntary, and individuals could choose whether to participate. Completed questionnaires and clinical data were coded by occupational health personnel at each participating company. All data remained anonymous and strictly confidential. This study followed the Declaration of Helsinki guidelines and was approved by the Institutional Review Board of Changhua Christian Hospital, Taiwan, with a waiver of informed consent (Approval No. CCH IRB 191238).

### 2.2. Study Design and Participants

A cross-sectional study using convenience sampling was conducted. Employees from four industrial enterprises in central Taiwan, aged 20 years or older, were invited to participate. The primary industries included the manufacturing of pumps, auto parts, electronic components, and transportation equipment. The businesses were selected due to their positive collaboration with the Center for Occupational Health, which facilitated the smooth execution of this study. In total, 4804 workers volunteered to participate.

In Taiwan, the proportion of women in the workforce is lower than in countries like the United States, resulting in a higher male-to-female ratio in this study. A chi-square goodness-of-fit test was performed to assess the gender distribution among participants: male (*n* = 3948, 82.2%) and female (*n* = 856, 17.8%) in the full sample, and male (*n* = 2571, 83.2%) and female (*n* = 518, 16.8%) in a subset of 3089 workers. The *p*-value of 0.134 indicated no statistically significant difference, suggesting that the sample data aligned with the population distribution of the four companies.

### 2.3. Assessment of Lifestyle Habits

Lifestyle habits were assessed using a self-administered questionnaire, focusing on dietary habits, physical activity, smoking, and alcohol consumption.

Dietary and physical activity habits were evaluated using subsets of the Health Promoting Lifestyle Profile II [17,18]. The nutrition behavior category included nine items, and the exercise behavior category consisted of eight items. Both categories were assessed using a four-point Likert scale, where possible responses included “Never”, “Sometimes”, “Often”, and “Routinely”. The average score for each subscale was calculated by dividing the total score by the number of items in that subscale. Higher scores indicated greater engagement in health-promoting behaviors. The dietary behavior items included questions about selecting a low-fat diet, restricting sugar consumption, and consuming adequate amounts of fruits, vegetables, and other food groups. The exercise behavior checklist included items related to adherence to an exercise program, participation in physical activities, performing stretching exercises, and monitoring heart rate during exercise. These items were translated into traditional Chinese to ensure participants’ comprehension. The results of this study indicated that the internal consistency of the nutrition and exercise behavior subscales within the Taiwanese population was satisfactory, with Cronbach’s alpha values of 0.78 and 0.85, respectively.

The smoking status of each participant was categorized as either a non-smoker or a current smoker (including both occasional and daily smokers). Alcohol consumption was categorized into two groups: non-drinkers and current drinkers (including both occasional and daily drinkers).

### 2.4. Evaluation of FL

The presence of FL was assessed using ultrasound imaging, which has been validated to reliably detect ≥5% histologically defined hepatic steatosis, thereby enhancing diagnostic accuracy and outperforming anthropometric and biochemical surrogate markers [19,20,21]. Based on the ultrasound findings, participants were categorized into two groups: without FL (Grade 0), characterized by a normal liver echotexture comparable to the cortex of the right kidney, and with FL (Grades 1–3), which included any degree of increased liver echogenicity, such as mild, moderate, or severe changes. The “with FL” group was identified by features such as impaired visualization of the portal vein wall, diaphragm, or posterior segment of the right liver lobe [19,20,21].

### 2.5. Liver Function Analysis

Liver function parameters were evaluated by measuring the levels of glutamate oxaloacetate transaminase (GOT) and glutamate pyruvate transaminase (GPT). The reference range for GPT was 0–35 U/L, while the reference range for GOT was 5–40 U/L. These reference values were used to interpret the results and assess liver function abnormalities.

### 2.6. Anthropometric and Biochemical Measurements

Waist circumference (WC) was measured at the midpoint between the lowest rib and the iliac crest using a non-stretchable tape while participants maintained an upright posture. To ensure consistency and accuracy, trained personnel performed all measurements. Blood pressure was assessed using a validated digital sphygmomanometer (HEM-7310, Omron, Kyoto, Japan) following the European Society of Hypertension guidelines. Systolic blood pressure (SBP) and diastolic blood pressure (DBP) were measured in a seated position after a 5 min rest. If the measured blood pressure values showed a significant deviation from the participant’s usual readings, an additional measurement was performed to ensure accuracy.

Participants were instructed to fast for 8–10 h before blood sample collection to ensure accurate biochemical analysis. Serum uric acid (UA), fasting blood glucose (FBG), and lipid profiles—including triglycerides (TG) and high-density lipoprotein cholesterol (HDL-C)—were measured using a Dimension RxL Max integrated clinical chemistry autoanalyzer (Siemens Healthcare Diagnostics, Deerfield, IL, USA). Biochemical parameters were analyzed using standardized enzymatic methods. FBG and TG levels were assessed via enzymatic colorimetric assays, while serum UA concentrations were quantified using the uricase–peroxidase method.

Hyperuricemia is defined as ≥7.0 mg/dL in men and ≥6.0 mg/dL in females [2]. Metabolic syndrome (MS) was diagnosed based on the criteria provided by the Health Promotion Administration, Ministry of Health and Welfare, Taiwan [22]. A diagnosis required the presence of at least three of the following components: (1) central obesity, defined as a WC ≥ 90 cm for men or ≥80 cm for women; (2) elevated blood pressure, indicated by SBP ≥ 130 mmHg or DBP ≥ 85 mmHg; (3) elevated FBG, with levels ≥ 100 mg/dL; (4) elevated TG, defined as levels ≥ 150 mg/dL; and (5) low HDL-C levels, identified as <40 mg/dL in men or <50 mg/dL in women. These criteria ensured a standardized approach to diagnosing MS within the study population.

### 2.7. Detection of Inflammatory Markers

White blood cell (WBC) counts were measured using direct current detection technology (XT1800i, Sysmex, Japan) to assess inflammatory status and cardiovascular risk [23]. WBC levels were measured using direct current detection technology (XT1800i, Sysmex, Kobe-shi, Hyogo, Japan), ensuring precision and reliability. Participants were stratified into three groups (low, medium, high) based on WBC tertiles. Elevated WBC levels (≥7.17 × 10^9^/L) indicated increased inflammatory and cardiovascular risk.

### 2.8. Statistical Analysis

Descriptive statistics were used to examine the basic characteristics of the data. The assessment of normality, conducted using the Kolmogorov–Smirnov test, skewness, and kurtosis, revealed that several variables were not normally distributed. Given this study’s emphasis on clinical significance, particularly in understanding the associations between hyperuricemia, FL, and metabolic abnormalities, inflammation, and liver dysfunction among workers, continuous variables were categorized based on established clinical definitions or various cut-off points.

According to the central limit theorem, the sampling distribution of the mean tends to approximate normality in large samples, such as in our study. Therefore, using the arithmetic mean rather than the median provides greater statistical power when comparing central tendencies between samples. Furthermore, both *t*-tests and ANOVA are generally robust to violations of normality. Independent samples *t*-tests were used to compare means between two groups, while comparisons among four groups were performed using one-way ANOVA with Scheffé’s post hoc analysis to identify specific differences, and polynomial contrasts were applied to evaluate linear trends across groups. The chi-square test was employed to examine associations between categorical variables, and logistic regression models were used to analyze the relationship between combined hyperuricemia and FL status (as the independent variable) and metabolic risk factors, inflammation, and liver dysfunction (as dependent variables). Odds ratios (95% confidence intervals) were calculated, with statistical significance defined as *p* < 0.05. All statistical analyses were conducted using IBM SPSS version 22.

## 3. Results

### 3.1. Baseline Characteristics of the Participants by Gender

As presented in Table 1, this study included 3089 participants, consisting of 2571 males and 518 females. The mean age of participants was 42.3 ± 10.0 years. Males had a significantly younger average age (41.6 ± 10.2 years) than females (45.3 ± 7.8 years, *p* < 0.001).

Regarding nutritional health behavior, females scored significantly higher than males (males: 2.42 ± 0.43, females: 2.65 ± 0.43, *p* < 0.001). However, no significant gender differences were observed in exercise-related health behavior scores. Smoking prevalence was markedly higher among males (27.3%) than females (1.0%, *p* < 0.001). Similarly, alcohol consumption was significantly more common in males (50.5%) than in females (19.3%, *p* < 0.001).

The overall prevalence of fatty liver (FL) was 43.2%, higher than that of hyperuricemia (25.5%). Hyperuricemia was significantly more common in males (28.2%) than in females (11.8%, *p* < 0.001). Similarly, FL prevalence was notably higher in males (46.6%) compared to females (26.4%, *p* < 0.001).

### 3.2. Differences in Health-Related Lifestyle Habits and Metabolic Risk Factors by Hyperuricemia and FL Status

As shown in Table 2, mean age did not differ significantly between the hyperuricemia and non-hyperuricemia groups (*p* = 0.468). However, participants with FL were significantly older than those without FL (43.0 ± 9.7 vs. 41.7 ± 10.2 years, *p* = 0.001).

Exercise-related health behaviors showed no significant differences between the hyperuricemia groups, but were significantly lower in participants with FL (1.91 ± 0.52 vs. 1.98 ± 0.59, *p* < 0.001). Nutritional health behavior scores did not differ across groups.

Smoking prevalence was similar between hyperuricemia groups but slightly higher in the FL group, showing a trend toward significance (*p* = 0.058). Alcohol consumption was significantly higher in participants with hyperuricemia (51.2% vs. 43.3%, *p* < 0.001) and FL (48.1% vs. 43.2%, *p* = 0.006).

Both hyperuricemia and FL were significantly associated with higher WC, SBP, DBP, TG, and lower HDL-C levels (all *p* < 0.001). FBG levels were also significantly higher in the FL group (*p* < 0.001).

For inflammatory markers and liver function, both conditions were linked to higher WBC counts, GPT, and GOT levels (all *p* < 0.001). Additionally, hyperuricemia prevalence was significantly higher in participants with FL (*p* < 0.001).

### 3.3. Differences in Lifestyle Habits, Metabolic Risk Factors, WBC Count, and Liver Function Indicators Among Groups from Combining Hyperuricemia with FL Status

As shown in Table 3, age progressively increased across the four groups, from Group A (no hyperuricemia/no FL) to Group D (hyperuricemia/FL, *p* = 0.009). The proportion of males was highest in Group D (93.9%) and lowest in females (6.1%, *p* < 0.001).

Exercise-related health behavior scores were highest in Group A (1.98 ± 0.59) and lowest in Groups C and D (1.91 ± 0.52, *p* = 0.007). No significant differences were observed in nutritional health behavior scores among the four groups (*p* = 0.329). Alcohol consumption was most prevalent in Group D (54.3%, *p* < 0.001).

A stepwise increase in WC, FBG, blood pressure (SBP, DBP), and TG was observed across Groups A to D, while HDL-C levels progressively decreased (all *p* < 0.001).

WBC counts, GPT, and GOT levels were highest in Group D (all *p* < 0.001). The prevalence of metabolic syndrome was also significantly higher in Group D (31.7%, *p* < 0.001), alongside the highest percentage of elevated WBC counts (49.2%, *p* < 0.001).

### 3.4. Odds Ratio of Metabolic Risk Factors, High WBC, and Liver Dysfunction Predicted by Combining Hyperuricemia with FL Status

Logistic regression models were used to assess the associations between hyperuricemia combined with FL (predictor) and various outcomes, including metabolic risk factors, elevated WBC count, and liver dysfunction (dependent variables), with adjustments for sex, age, and lifestyle factors.

Table 4 compares metabolic risk factors across four groups, using Group A (no hyperuricemia/no FL, OR = 1.00) as the reference. Risk increased progressively, with Group B (hyperuricemia/no FL) showing moderate elevation, Group C (no hyperuricemia/FL) a more significant rise, and Group D (hyperuricemia/FL) the highest risk, highlighting their combined detrimental effects. Most indicators followed the pattern “A < B < C < D” or “A < B ≒ C < D”.

For central obesity, ORs were 1.98 (Group B), 6.87 (Group C), and 12.39 (Group D), following “A < B < C < D“. Elevated FPG showed no significant difference in Group B (OR = 0.96, *p* = 0.795) but increased in Group C (OR = 1.72) and Group D (OR = 2.24), following “A ≒ B < C < D”.

Elevated TG risk was 2.42 (Group B), 2.95 (Group C), and 5.69 (Group D), following “A < B ≒ C < D”. Similarly, low HDL-C risk increased from 1.80 (Group B) to 3.35 (Group C) and 4.30 (Group D), following “A < B < C ≤ D”.

For elevated blood pressure, ORs were 1.77 (Group B), 1.98 (Group C), and 3.06 (Group D), while MS followed a steep increase: 2.90 (Group B), 6.15 (Group C), and 11.52 (Group D), following “A < B < C < D”.

Inflammatory markers showed no significant difference in WBC count between Groups A and B (OR = 1.29) but increased in Group C (OR = 1.58) and Group D (OR = 2.91), following “A ≤ B ≒ C < D”.

For liver function, elevated GPT risk was 1.67 (Group B), 3.16 (Group C), and 5.45 (Group D), following “A < B < C < D”. Elevated GOT showed no significant difference in Group B (OR = 1.65) but increased in Group C (OR = 1.65) and Group D (OR = 3.14), following “A ≒ B ≒ C < D”.

In summary, Table 4 reveals that the combined presence of hyperuricemia and FL significantly worsens metabolic, inflammatory, and liver function abnormalities, with Group D facing the highest risks.

## 4. Discussion

This cross-sectional study investigated the relationships between lifestyle habits, FL, and hyperuricemia among workers, emphasizing key health risk factors. The findings revealed that: (1) FL (43.2%) was more prevalent than hyperuricemia (25.5%), with males significantly more affected. (2) Individuals with FL and hyperuricemia exhibited harmful lifestyle choices. (3) Both hyperuricemia and FL were strongly associated with increased metabolic risk factors, liver dysfunction, and elevated inflammatory markers. (4) The highest odds of MS, inflammation, and liver dysfunction were seen in those who had both FL and hyperuricemia.

### 4.1. Relationship Between Hyperuricemia, FL, and Gender

The prevalence of hyperuricemia has shown a significant upward trend worldwide, with rates varying from 2.6% to 36% across different populations [2,24]. In Taiwan, the prevalence of hyperuricemia was 13.8% and remains a pressing public health concern, affecting 17.9% of males and 9.9% of females [25]. Similarly, FL disease has emerged as the leading cause of liver disease globally. Its prevalence has steadily increased, with current estimates indicating that 32% of adults worldwide are affected, with a notably higher prevalence in males (40%) compared to females (26%) [26]. The prevalence rate of FL in Taiwan is approximately 33.3% [27]. It is evident that Taiwan must address the problems of FL and hyperuricemia.

FL disease and hyperuricemia exhibit significant gender disparities, with males being disproportionately affected. In this study, the percentage of FL (43.2%) and hyperuricemia (25.5%) was higher among male participants compared to females. A study on Taiwanese police officers also showed gender differences, with 52.2% found to have nonalcoholic FL disease, with men having a higher prevalence than women (53.6% vs. 20.5%) [28]. This observation aligns with the present study and with global and regional trends reported in previous studies, which emphasize significant gender disparities in the prevalence and progression of these conditions.

These gender differences may be attributed to biological factors, such as variations in sex hormones, differences in body fat distribution, and lifestyle behaviors, including dietary patterns and alcohol consumption [29,30]. This study shows that men have worse nutritional health behaviors than women, and men smoke and drink at higher rates than women. Additionally, male nonalcoholic FL disease patients with hyperuricemia have been reported to experience more frequent and severe liver injury compared to their female counterparts [31].

These gender-specific patterns underscore the need for tailored prevention strategies that address the unique risk profiles of males and females. Future research should further explore the interplay of hormonal, genetic, and lifestyle factors in gender-related disparities to inform targeted interventions.

### 4.2. Lifestyle Habits According to Hyperuricemia and FL Status

This study showed that FL (43.2%) was more prevalent than hyperuricemia (25.5%) in workers. The higher prevalence of FL compared to hyperuricemia in workers is likely due to FL disease being more directly linked to MS components, such as obesity, insulin resistance, and dyslipidemia, which are prevalent among workers due to occupational stress, sedentary behavior, and poor diet [7,9,32,33,34]. Hyperuricemia, while also linked to these conditions [35], tends to have a lower overall prevalence.

Lifestyle factors are closely associated with the development of non-alcoholic FL disease [3,36]. The present study indicated that participants with FL exhibited distinct lifestyle and metabolic risk profiles compared to those without. Lower exercise-related health behavior scores and higher alcohol consumption were notable among individuals with FL, highlighting the critical role of modifiable lifestyle factors. Sedentary lifestyles and excessive alcohol consumption are well-established risk factors for FL [37]. Recent studies emphasize that regular physical activity can mitigate FL risk by improving insulin sensitivity and reducing intrahepatic fat [38,39].

Similarly, this study showed that alcohol consumption was significantly higher in the hyperuricemia group compared to the non-hyperuricemia group. According to a cross-sectional study of male office workers in Japan, drinking alcohol is linked to a higher risk of hyperuricemia, and the risk is the same regardless of the type of alcoholic beverage consumed [34]. The relationship between alcohol consumption and hyperuricemia has been widely recognized [40,41]. Alcohol intake is generally associated with elevated serum uric acid levels, primarily due to the acceleration of purine metabolism and the reduced renal clearance of uric acid [40,42,43]. Furthermore, excessive alcohol consumption exacerbates liver damage by increasing oxidative stress and fat accumulation, as evidenced in prior research [44,45].

Individuals with both hyperuricemia and FL displayed the most detrimental lifestyle habits, including insufficient physical activity, excessive alcohol consumption, and poor adherence to health-promoting behaviors. These findings highlight the need for targeted interventions focusing on lifestyle modifications to reduce metabolic risks and improve overall health.

### 4.3. Metabolic Risk Factors, Inflammatory Markers, and Liver Function in Hyperuricemia and FL

This study showed that hyperuricemia and FL were both associated with higher WC, SBP, DBP, TG, and lower HDL-C. FL disease is often closely related to insulin resistance, which causes excessive secretion of TG by the liver, increases TG levels in serum, and promotes the accumulation of visceral fat, leading to an increase in WC [46,47]. At the same time, insulin resistance can lead to hypertension because insulin affects vasomotor function [47]. Additionally, FL-related lipid metabolism disorders often result in hypertriglyceridemia and low HDL-C [47].

Hyperuricemia is associated with metabolic abnormalities, including oxidative stress and purine metabolism dysregulation [2]. While UA has antioxidant properties at low concentrations, elevated levels induce oxidative stress, impair endothelial function, and increase hypertension risk [2]. UA also promotes adipose inflammation, exacerbating insulin resistance and lipid abnormalities [2]. Moreover, hyperuricemia is prevalent in MS and strongly linked to obesity, elevated TG, and reduced HDL-C [2].

Both hyperuricemia and FL disease may involve chronic low-grade inflammation. This study found that individuals with hyperuricemia and FL had higher WBC counts, GPT, and GOT levels. Uric acid crystals and pro-inflammatory factors (such as TNF-α and IL-6) released by adipocytes can activate the immune system, causing WBC counts to rise [2]. In FL, liver cells are damaged by excessive lipid accumulation and release GPT and GOT [48]. Hyperuricemia further aggravates liver damage by promoting oxidative stress and inflammation in the liver [49,50]. This may also explain why this study found that the proportion of hyperuricemia in the FL group was significantly higher than that in the non-FL group.

### 4.4. Relationship Between Combined Hyperuricemia and Fatty Liver and Health Risk Profiles

By encouraging oxidative stress and inflammation in the liver, hyperuricemia exacerbates liver damage [49,50]. FL disease linked to metabolic dysfunction and hyperuricemia may create a vicious cycle that worsens metabolic state [51]. Therefore, combining hyperuricemia and FL status may reveal a compounding effect on metabolic risk factors and health outcomes.

The present study showed a progressive increase in age, poor health behaviors, metabolic risks, inflammatory markers, and liver dysfunction indicators from Group A to Group D (hyperuricemia/FL). Group D showed the highest prevalence of males, alcohol consumption, MS, and elevated WBC counts. Exercise-related health behavior scores were lowest in Groups C and D, while nutritional scores showed no significant differences.

In addition, the odds ratio analysis revealed central obesity risk increased from Group A to Group D, with the highest odds in Group D. Elevated FBG and high WBC levels were higher in Groups C and D, while liver dysfunction risks progressively increased across Groups B, C, and D. Overall, individuals with both FL and hyperuricemia exhibit significantly increased odds of metabolic dysfunction, suggesting an additive—or potentially synergistic—cardiometabolic burden. This finding aligns with prior evidence that both hepatic steatosis and hyperuricemia, independently and in combination, contribute to insulin resistance, systemic inflammation, and atherogenesis [2,46,47,48,49,50,51]. The findings of this study suggest two key implications: (1) FL and hyperuricemia appear to have synergistic effects, intensifying metabolic abnormalities and increasing the risk of liver damage. Persons with both conditions necessitate more rigorous and comprehensive health management strategies. (2) Lifestyle modifications, such as promoting regular physical activity and reducing alcohol consumption, are vital for preventing and managing hyperuricemia and FL disease. Focused interventions tailored to high-risk populations (e.g., Groups C and D) are particularly crucial for effective disease control and prevention.

Overall, this study underscores the synergistic relationship between hyperuricemia and FL in worsening metabolic health. Future interventions should prioritize early detection and holistic management of metabolic risk factors and lifestyle behaviors to mitigate long-term health impacts. Taiwan’s national health system has implemented preventive programs targeting metabolic syndrome, including community-based health screenings and lifestyle modification initiatives. Our findings underscore the potential value of incorporating fatty liver and serum uric acid screening into occupational health evaluations to better identify high-risk individuals and align with these preventive strategies.

### 4.5. Achievements and Implications

This study provides valuable insights into the relationship between FL, hyperuricemia, and lifestyle habits among workers, emphasizing their role as health risk factors. The findings reveal a synergistic effect of hyperuricemia and FL in worsening metabolic disorders and highlight the high prevalence of both conditions, with FL being more common. This underscores the need for comprehensive health promotion strategies at both individual and organizational levels. Employers can support workers’ health by implementing wellness programs that promote a healthy lifestyle, such as avoiding excessive alcohol consumption, regular exercise, and routine screenings. Additionally, workplace policies that reduce stress and enhance work–life balance may help mitigate behavioral risks, improving both health and productivity.

### 4.6. Limitations and Prospective

Despite its strengths, this study has several limitations. The cross-sectional design precludes causal inference, and the reliance on self-reported data may introduce reporting bias. Future research should employ longitudinal designs to establish causality and explore the long-term outcomes of interventions targeting fatty liver and hyperuricemia. Incorporating objective measures of dietary and physical activity patterns could enhance the accuracy of lifestyle assessments. Moreover, further studies in other employee populations and occupational settings are warranted to validate the generalizability of the findings. Variations in age and lifestyle characteristics across groups may also have contributed to the observed associations. Nonetheless, statistical adjustments were performed to control for and mitigate the potential influence of these confounding variables.

## 5. Conclusions

This study examined the combined effects of FL and hyperuricemia on metabolic risk factors, inflammation, and abnormal liver function in Taiwanese workers. Our findings highlight significant gender differences, the adverse impact of poor lifestyle habits, and the synergistic effect of hyperuricemia and FL in worsening metabolic disorders, inflammation, and liver dysfunction. These results emphasize the need for targeted prevention and management strategies, particularly for high-risk populations. Future research should explore underlying mechanisms and assess integrated interventions to mitigate these health risks.

## Figures and Tables

**Table 1 metabolites-15-00318-t001:** Baseline characteristics of the participants by gender.

Variables	Total (N = 3089)	Gender		*p*
		Male (*n* = 2571)	Female (*n* = 518)	
Age (year)	42.3 ± 10.0	41.6 ± 10.2	45.3 ± 7.8	<0.001
Lifestyle habits				
Exercise health behavior	1.95 ± 0.56	1.95 ± 0.56	1.98 ± 0.55	0.228
Nutritional health behavior	2.45 ± 0.44	2.42 ± 0.43	2.65 ± 0.43	<0.001
Smoking				
With	706 (22.9)	701 (27.3)	5 (1.0)	<0.001
Without	2383 (77.1)	1870 (72.7)	513 (99.0)	
Alcohol consumption				
With	1399 (45.3)	1299 (50.5)	100 (19.3)	<0.001
Without	1690 (54.7)	1272 (49.5)	418 (80.7)	
Hyperuricemia				
With	787 (25.5)	726 (28.2)	61 (11.8)	<0.001
Without	2302 (74.5)	1845 (71.8)	457 (88.2)	
FL				
With	1335 (43.2)	1198 (46.6)	137 (26.4)	<0.001
Without	1754 (56.8)	1373 (53.4)	381 (73.6)	

Abbreviations: FL, fatty liver. The means of two groups were compared using *t*-test. Data are means ± SD. Statistically significant at *p* < 0.05. The chi-square test was used to examine association between two categorical variables. Data are number (n), percent (%). Significant difference (*p* < 0.05).

**Table 2 metabolites-15-00318-t002:** Differences in health-related lifestyle habits and metabolic risk factors by hyperuricemia and FL status.

Variables	Hyperuricemia		*p*	FL		*p*
	With (*n* = 787)	Without (*n* = 2302)		With (*n* = 1335)	Without (*n* = 1754)	
Age (year)	42.5 ± 10.1	42.2 ± 9.9	0.468	43.0 ± 9.7	41.7 ± 10.2	0.001
Lifestyle habits						
Exercise health behavior	1.95 ± 0.54	1.95 ± 0.57	0.773	1.91 ± 0.52	1.98 ± 0.59	<0.001
Nutritional health behavior	2.45 ± 0.44	2.45 ± 0.44	0.997	2.44 ± 0.42	2.47 ± 0.45	0.098
Smoking						
With	190 (24.1)	519 (22.4)	0.319	327 (24.5)	379 (21.6)	0.058
Without	597 (75.9)	1786 (77.6)		1008 (75.5)	1375 (78.4)	
Alcohol consumption						
With	403 (51.2)	996 (43.3)	<0.001	642 (48.1)	757 (43.2)	0.006
Without	384 (48.8)	1306 (56.7)		693 (51.9)	997 (56.8)	
Metabolic risk factors						
WC (cm)	86.2 ± 9.1	80.3 ± 9.0	<0.001	87.1 ± 8.3	77.8 ± 8.1	<0.001
FBG (g/dL)	93.9 ± 14.3	93.2 ± 20.2	0.405	96.7 ± 22.0	90.9 ± 15.6	<0.001
SBP (mmHg)	127.0 ± 15.2	121.2 ± 14.9	<0.001	125.9 ± 15.3	120.2 ± 14.7	<0.001
DBP (mmHg)	81.6 ± 11.4	77.2 ± 10.9	<0.001	80.6 ± 11.5	76.6 ± 10.6	<0.001
Triglycerides (mg/dL)	167.2 ± 119.1	122.9 ± 97.2	<0.001	163.3 ± 119.2	112.0 ± 86.4	<0.001
HDL-C (mg/dL)	49.4 ± 11.94	53.7 ± 12.7	<0.001	48.2 ± 11.1	56.0 ± 12.7	<0.001
Inflammatory marker						
WBC count (10^9^/L)	7.1 ± 2.1	6.6 ± 1.7	<0.001	7.0 ± 1.6	6.4 ± 1.9	<0.001
Liver function indicators						
GPT (U/L)	33.2 ± 29.6	24.2 ± 21.7	<0.001	32.8 ± 25.4	21.7 ± 22.2	<0.001
GOT (U/L)	23.8 ± 15.4	20.3 ± 19.4	<0.001	22.4 ± 11.0	20.2 ± 22.5	0.001
Metabolic syndrome						
With	179 (22.7)	234 (10.2)	<0.001	324 (24.3)	89 (5.1)	<0.001
Without	608 (77.3)	2068 (89.8)		1011 (75.7)	1665 (94.9)	
Inflammatory marker						
WBC (10^9^ cells/L)						
Low ≤ 5.87	184 (23.4)	851 (37.0)	<0.001	300 (22.5)	735 (41.9)	<0.001
Med. 5.88–7.16	272 (34.6)	759 (33.3)		495 (37.1)	536 (30.6)	
High ≥ 7.17	331 (42.1)	692 (30.1)		540 (40.4)	483 (27.5)	
FL						
With	457 (58.1)	878 (38.1)	<0.001			
Without	330 (41.9)	1424 (61.9)				
Hyperuricemia						
With				457 (34.2)	330 (18.8)	<0.001
Without				878 (65.8)	1424 (81.2)	

Abbreviations: WC, waist circumference; FBG, fasting blood glucose; SBP, systolic blood pressure; DBP, diastolic blood pressure; HDL-C, high-density lipoprotein cholesterol; WBC, white blood cell; GOT, glutamate oxaloacetate transaminase; GPT, glutamate pyruvate transaminase; FL, fatty liver. The means of two groups were compared using *t*-test. Data are means ± SD. Statistically significant at *p* < 0.05. The chi-square test was used to examine association between two categorical variables. Data are number (n), percent (%). Significant difference (*p* < 0.05).

**Table 3 metabolites-15-00318-t003:** The differences in lifestyle habits, metabolic risk factors, WBC count, and liver function indicators by combining hyperuricemia with fatty liver status.

Variables	Group A	Group B	Group C	Group D	*p* for ANOVA or Chi-Square	*p* for Linear Trend
	(Without Hyperuricemia/Without FL) (*n* = 1424)	(With Hyperuricemia/ Without FL) (*n* = 330)	(Without Hyperuricemia/ with FL) (*n* = 878)	(With Hyperuricemia/ with FL) (*n* = 457)		
Percent (%)	46.1	10.7	28.4	14.8		
Age (year) ^b^	41.7 ± 10.0	41.7 ± 10.7	42.9 ± 9.7	43.0 ± 9.7	0.009	0.003
Gender						
Male	1076 (75.6)	297 (90.0)	769 (87.6)	429 (93.9)	<0.001	<0.001
Female	348 (24.4)	33 (10.0)	109 (12.4)	28 (6.1)		
Lifestyle habits						
Exercise health behavior ^b^	1.98 ± 0.59	2.00 ± 0.55	1.91 ± 0.52	1.91 ± 0.53	0.007	0.003
Nutritional health behavior	2.47 ± 0.45	2.46 ± 0.45	2.43 ± 0.41	2.45 ± 0.42	0.329	0.843
Smoking						
With	297 (20.9)	82 (24.8)	219 (24.9)	108 (23.6)	0.098	0.045
Without	1127 (79.1)	248 (75.2)	659 (75.1)	349 (76.4)		
Alcohol consumption						
With	602 (42.3)	155 (47.0)	394 (44.9)	248 (54.3)	<0.001	<0.001
Without	822 (57.7)	175 (53.0)	484 (55.1)	209 (45.7)		
Metabolic risk factors						
WC (cm) ^a,b,c,d,e,f^	76.9 ± 7.9	81.6 ± 8.0	85.8 ± 8.0	89.5 ± 8.4	<0.001	<0.001
FBG (g/dL) ^b,c,d,e^	90.9 ± 16.4	90.6 ± 11.4	96.9 ± 24.7	96.2 ± 15.7	<0.001	<0.001
SBP (mmHg) ^a,b,c,e,f^	119.2 ± 14.5	124.6 ± 14.6	124.3 ± 15.0	128.8 ± 15.4	<0.001	<0.001
DBP (mmHg) ^a,b,c,e,f^	75.8 ± 10.3	80.0 ± 11.3	79.4 ± 11.3	82.8 ± 11.4	<0.001	<0.001
Triglycerides (mg/dL) ^a,b,c,e,f^	105.1 ± 76.4	141.9 ± 115.8	151.7 ± 118.1	185.5 ± 118.2	<0.001	<0.001
HDL-C (mg/dL) ^a,b,c,d,e^	56.8 ± 12.7	52.6 ± 12.1	48.8 ± 11.0	47.2 ± 11.3	<0.001	<0.001
Inflammatory marker						
WBC count (10^9^/L) ^a,b,c,e,f^	6.4 ± 1.7	6.8 ± 2.6	6.9 ± 1.6	7.3 ± 1.6	<0.001	<0.001
Liver function indicators						
GPT (U/L) ^a,b,c,e,f^	20.7 ± 20.1	26.1 ± 29.1	29.9 ± 23.0	38.2 ± 28.9	<0.001	<0.001
GOT (U/L) ^c,f^	19.7 ± 23.3	22.5 ± 18.9	21.2 ± 10.1	24.7 ± 12.2	<0.001	<0.001
MS						
With	55 (3.9)	34 (10.3)	179 (20.4)	145 (31.7)	<0.001	<0.001
Without	1369 (96.1)	296 (89.7)	699 (79.6)	312 (68.3)		
High WBC ≥ 7.17 (10^9^ cells/L)						
With	377 (26.5)	106 (32.1)	315 (35.9)	225 (49.2)	<0.001	<0.001
Without	1047 (73.5)	224 (67.9)	563 (64.1)	232 (50.8)		

Abbreviations: WC, waist circumference; FBG, fasting blood glucose; SBP, systolic blood pressure; DBP, diastolic blood pressure; HDL-C, high-density lipoprotein cholesterol; MS, metabolic syndrome; WBC, white blood cell; GOT, glutamate oxaloacetate transaminase; GPT, glutamate pyruvate transaminase; FL, fatty liver. One-way ANOVA with Scheffe’s post hoc test was used to compare age, exercise and nutrition, metabolic risk factors, WBC count, and liver function across hyperuricemia and FL status groups. ANOVA trend analysis was conducted using polynomial contrasts. Data are presented as mean ± SD. Multiple comparisons: a = A vs. B, b = A vs. C, c = A vs. D, d = B vs. C, e = B vs. D, f = C vs. D. Significant difference (*p* < 0.05). Chi-square test was used to assess associations between categorical variables. The Cochran–Armitage test was applied for trend analysis (linear-by-linear association). Data are presented as n (%). Significant difference (*p* < 0.05).

**Table 4 metabolites-15-00318-t004:** Odds ratios of metabolic risk factors, high WBC, and liver dysfunction by combining hyperuricemia with fatty liver status.

Variables	Group A	Group B	Group C	Group D	Odds Ratio Comparison Between Four Groups
	(Without Hyperuricemia /Without FL) (*n* = 1424)	(With Hyperuricemia /Without FL) (*n* = 330)	(Without Hyperuricemia /with FL) (*n* = 878)	(With Hyperuricemia /with FL) (*n* = 457)	
Metabolic risk factors					
WC (cm) ≥90 for men or ≥80 for women (central obesity)	1.00	1.98 (1.35–2.91)	6.87 (5.35–8.82)	12.39 (9.35–16.41)	A < B < C < D
*p*		0.001	<0.001	<0.001	
FPG ≥ 100 mg/dL	1.00	0.96 (0.67–1.35)	1.72 (1.38–2.15)	2.24 (1.73–2.90)	A ≒ B < C < D
*p*		0.795	<0.001	<0.001	
Triglycerides ≥ 150 mg/dL	1.00	2.42 (1.82–3.20)	2.95 (2.41–3.63)	5.69 (4.47–7.24)	A < B ≒ C < D
*p*		<0.001	<0.001	<0.001	
HDL-C (mg/dL) <40 for men or <50 for women	1.00	1.80 (1.24–2.61)	3.35 (2.61–4.30)	4.30 (3.20–5.77)	A < B < C ≤ D
*p*		0.002	<0.001	<0.001	
SBP ≥ 130 and DBP ≥ 85	1.00	1.77 (1.37–2.29)	1.98 (1.65–2.38)	3.06 (2.44–3.85)	A < B ≒ C < D
*p*		<0.001	<0.001	<0.001	
Metabolic syndrome	1.00	2.90 (1.85–4.56)	6.15 (4.46–8.46)	11.52 (8.17–16.23)	A < B < C < D
*p*		<0.001	<0.001	<0.001	
Inflammatory marker					
High WBC ≥ 7.17 10^9^ cells/L	1.00	1.29 (0.98–1.69)	1.58 (1.31–1.91)	2.91 (2.31–3.67)	A ≤ B ≒ C < D
*p*		0.067	<0.001	<0.001	
Liver dysfunction					
GPT > 35 U/L	1.00	1.67 (1.18–2.37)	3.16 (2.48–4.03)	5.45 (4.15–7.18)	A < B < C < D
*p*		0.004	<0.001	<0.001	
GOT > 40 U/L	1.00	1.65 (0.85–3.19)	1.65 (1.01–2.69)	3.14 (1.89–5.20)	A ≒ B ≒ C < D
*p*		0.137	0.047	<0.001	

Abbreviations: WC, waist circumference; FPG, fasting plasma glucose; SBP, systolic blood pressure; DBP, diastolic blood pressure; HDL-C, high-density lipoprotein cholesterol; WBC, white blood cell; GOT, glutamate oxaloacetate transaminase; GPT, glutamate pyruvate transaminase; FL, fatty liver. Logistic regression models were used to assess the association between hyperuricemia combined with FL (predictor) and metabolic risk factors, elevated WBC, and liver dysfunction (dependent variables). Models were adjusted for sex, age, nutrition and exercise, smoking, and alcohol consumption. Results are presented as odds ratios (95% CIs), with statistical significance set at *p* < 0.05. Groups A, B, and C were used as reference categories to compare the odds ratios of metabolic risk factors, elevated WBC, and liver dysfunction, ranking the levels accordingly. <: Significantly lower odds ratio than another group. ≤: Marginally significantly lower odds ratio. ≒ No significant difference between groups.

## Data Availability

The datasets generated and analyzed during the current study are not publicly available due to ethical and privacy considerations.

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
