# Peer review of "Fatty Liver and Hyperuricemia in Workers: Combined Effects on Metabolic Dysfunction and the Role of Lifestyle Factors"

_metabolites, 2025, doi:10.3390/metabo15050318_

Round 1
Reviewer 1 Report
Comments and Suggestions for Authors
This study evaluating the combined effect of ultrasonographic fatty Liver and hyperuricemia on metabolic dysfunction manuscript is well-written and conclusions are supported by results. I have few comments aimed at improving the quality fo the manuscript. The use of US instead of anthropometric-serum surrogates is a point of strenght of the study given that US is much more accurate and allows for reliable detection of ≥ 5% histologically defined hepatic steatosi compared to histology (Metab Target Organ Damage 2021;1:7). Serum uric acid have been shown as an indepedent predictor of advanced NAFLD and its histological features (Hepatol Res. 2016 Oct;46(11):1074-1087.; Aliment Pharmacol Ther. 2011 Oct;34(7):757-66.). Patients with FL+/SUA + will probably display the highest cardiometabolic risk. Please discuss.
Reviewer 2 Report
Comments and Suggestions for Authors
The work submitted for review concerns socially important health problems related to metabolic disorders, i.e. fatty liver and hyperuricemia in the population of employees. An important advantage of the work is the study of a large population covering a total of 3089 participants.
The abstract should mention which group of employees the study concerns and from what geographical area the subjects come from.
In the methodology, the authors mentioned that the study was exempted from informed patient consent to participate in the study. Please provide the reasons for this decision. What is the justification for such a decision if the study was conducted on living, adult individuals?
The authors presented the results of the study and a discussion based on previous reports published by other authors. In the discussion part, they also included limitations related to the study. I have no objections to this part of the work.
Although the topic is well known, we need population studies to understand the scale of the problem and identify groups more vulnerable to metabolic diseases. I recommend the work for publication after minor improvements.
Minor errors:
- the abbreviation -p expressing the level of probability is sometimes written in italics and sometimes not. Please standardize.
- why were means compared in groups and not medians?
- it was shown that the compared groups differ statistically significantly in terms of age and lifestyle. Please mention this in the subsection on the limitations of the study.
- regarding future studies, the need to study other groups of employees should be mentioned.
- in the discussion, a subsection on current preventive programs for metabolic diseases should be added and reference should be made to them based on the obtained results.
